# Biochemical Pilot Study on Effects of Pomegranate Seed Oil Extract and Cosmetic Cream on Neurologically Mediated Skin Inflammation in Animals and Humans: A Comparative Observational Study

**DOI:** 10.3390/molecules28020903

**Published:** 2023-01-16

**Authors:** Asmaa Fathi Hamouda, Shifa Felemban

**Affiliations:** 1Department of Biochemistry, Faculty of Science, University of Alexandria, Alexandria 21111, Egypt; 2Department of Chemistry, Faculty of Applied Science, University College-Al Leith, University of Umm Al-Qura, Makkah 21955, Saudi Arabia

**Keywords:** collagenase, cosmetic, cyclooxygenase-2, elastase, formaldehyde, hyaluronidase, phenobarbital, skin inflammation, tyrosinase

## Abstract

The presence of phenobarbital and formaldehyde in drugs, food, and beverages can lead to various health issues, including inflammation, oncogenesis, and neurological distress. Psychological stress leads to mood fluctuations and the onset of skin inflammation. Skin inflammation has a range of causes, including chemicals, heavy metals, infection, immune-related disorders, genetics, and stress. The various treatments for skin inflammation include medical and cosmetic creams, diet changes, and herbal therapy. In this study, we investigated the effects of Avocom-M and pomegranate seed oil extract (PSOE) against phenobarbital- and formaldehyde-induced skin biochemical changes in rats. We analyzed the constituents of PSOE using gas chromatography–mass spectrometry and inductively coupled plasma–mass spectrometry. We also observed biochemical changes in the skin of human volunteers with and without TROSYD and PSOE as a skin cream. We compared the biochemical changes in human volunteers’ skin before treatment and 21 days after the treatment stopped. The outcomes showed an improvement in the rats’ biochemical status, due to PSOE and Avocom-M treatment. The human volunteers treated with TROSYD and PSOE showed substantial amelioration of skin inflammation. PSOE, Avocom-M, and TROSYD produced beneficial effects by reducing the levels of cyclooxygenase-2, lipid peroxidation, tyrosinase, hyaluronidase, elastase, collagenase, and nitric oxide in the animals tested on and in human volunteers.

## 1. Introduction

Skin is the first indicator in the body of internal and external emotional feelings, and the first line of defense against physical, chemical, and cosmetic challenges, heavy metals, and biological invaders [1,2]. Skin also has metabolic and endocrine roles in the body. These roles include the following: manufacturing of vitamin D, activation of constitutive steroid hormones, provision of sources of precursor compounds, and involvement in parathyroid hormone-related protein, catecholamines, and acetylcholine neurotransmitters, and with cytokinin that manages the biological role of the skin’s stress responses [3,4]. As such, understanding skin aging and the invention of new treatments for aging, infection, and skin diseases have become research hotspots for scientists [2,5].

Previous findings reported that L-tyrosine undergoing multistep transformations to yield melanin, including mixed melanin pigment, is a precursor to catecholamines [6,7]. The catecholamines have different oxidation–reduction reactions to provide neuromelanin. It was also reported that dopamine and cysteinyl dopamine can provide skin pigment [6,7]. Moreover, the previous publication revealed in detail that melanin pigment has a critical skin protection role against cancerogenesis, such as that of melanoma and melanogenesis that lead to increased proinflammatory mediators, reactive oxygen species (ROS), and peroxidation [8]. An imbalance between antioxidants and pro-oxidants creates ROS, as do various infections [9]. ROS initiate endogenous oxidative stress in the epidermis, inducing wrinkles, roughness, dryness, elasticity loss, inflammation, and uneven pigmentation changes [10].

Collagen, elastin, and hyaluronic acid are abundant and necessary components of the skin. Collagen provides the tensile strength of the skin, while elastin fibers are responsible for the elasticity of the skin [10]. ROS activate the synthesis of elastase and collagenase and increase premature skin aging, as evidenced by symptoms such as wrinkles, freckles, inflammation, and sallowness, among others [11]. Hyaluronic acid boosts skin renewal, controls moisture, elevates consistency, and decreases the permeability to extracellular fluid that occurs during the aging process or during skin infections [12].

Moreover, infection and the overproduction of ROS, which are associated with the overproduction of melanin, cause skin ailments and post-inflammatory hyperpigmentation, leading to the appearance of premature aging [13]. Tyrosinase is a rate-limiting factor in melanin pigmentation. Hence, the inhibition of tyrosinase activity may help to treat pigmentation and inflammation of the skin [2,13].

Nowadays, cosmetic products play an essential role in skin issues [14,15,16]. A cosmetic product such as Avocom-M Cream, Miconazole Nitrate 2% *w*/*w*, Mometasone Furoate 0.1%. (50 g, product code: 01053001) is described by a dermatologist as anti-inflammatory and antifungal. TROSYD, 1% Tioconazole Dermal Cream 50 g is a cosmetic cream described by dermatologists as antifungal and anti-bacterial [14,15,16].

Vitamins, lifestyle, a natural diet, herbs, natural products, and the use of natural cosmetic ingredients with high amounts of antioxidant composites are the best methods to manage skin inflammation and other disease conditions [17]. Pomegranate (*Punica granatum* L.) shows strong antioxidant, anti-inflammatory, anticancer, antibacterial, and anti-obesity activity [18]. As a powerful antioxidant, and immunomodulatory and anti-inflammatory agent, pomegranate can block oxidative stress. It can also decrease ROS, NOS activity, and NO production, caspase-3, DNA fragmentation, and apoptosis [18,19]. Additionally, pomegranate inhibits oxidation sensitivity, boosts the levels of the antioxidant enzymes glutathione, glutathione peroxidase, and catalase, diminishes immune cell count, and stimulates the proliferation of keratinocytes, which play a fundamental role in the immune response against infection. Moreover, pomegranate has been used to treat acute and chronic wounds, such as surgical wounds, burns, oral, stomatitis, and diabetic wounds, and gastric ulcers, producing results better than, or comparable to, those of commercial medicines, and with no adverse effects [19]. Recent studies have shown that pomegranate has antioxidant, anti-inflammatory, and antiapoptotic effects [18,20,21,22].

The chemical constituents of pomegranate seed oil extract (PSOE), as determined via gas chromatography–mass spectrometry (GC-MS) and inductively coupled plasma–mass spectrometry (ICP–MS) in our previous and current studies, have advantageous effects. The compounds in PSOE include octadecenamide, tocopherol, oleamide, squalene, stigmas-3,5-diene, and other potential bioactive phytochemical mixtures. Most of the compounds identified in PSOE show antiapoptotic, anti-inflammatory, antioxidant, anticancer, anti-obesity, antimicrobial, and antidiabetic activities [22].

In this study, we separately investigated the effects of treatment with PSOE and Avocom-M on phenobarbital- and formaldehyde-induced biochemical changes in a rat model, and conducted parallel observations of human volunteers that were administered TROSYD and PSOE separately as an intervention for skin inflammation issues. Animal models are widely used in biochemistry research, and the outcomes of animal studies usually form the foundation of pilot studies conducted with humans. From this viewpoint, our investigation resembles an animal pilot study performed in conjunction with the collection of notes on human responses to PSOE against a different kind of skin inflammation. We aimed to investigate the effect of PSOE and Avocom-M treatment on indicators of skin inflammation, such as the levels of cyclooxygenase-2, lipid peroxidation, tyrosinase, hyaluronidase, elastase, collagenase, and nitric oxide, induced by phenobarbital and formaldehyde in rats. We also separately studied the changes in the levels of these inflammatory indicators in human volunteers who received treatment with TROSYD and PSOE against inflammation and those without the treatment, and compared the results with those from the rat study. Figure 1 shows the experimental design [23,24,25,26].

## 2. Results

### 2.1. Metal Content Analysis of PSOE Using ICP–MS

Table 1 shows the metal contents in PSOE determined using ICP–MS.

### 2.2. GC-MS Analysis of PSOE

Table 2 shows the chemical composition of the PSOE, identified using GC–MS [22].

### 2.3. Biochemical Comparisons of Different Groups in Animal Skin Study

Figure 2 shows the results of the different parameters studied, reported as mean ± SD for eight rats. Figure 2 shows that phenobarbital and formaldehyde induced skin oxidative stress and inflammation in rats, which we recognized by significant (*p* ≤ 0.05) increases in the levels of collagenase (49.9 × 10^3^%), elastase (19.9 × 10^3^%), hyaluronidase (39.9 × 10^3^%), tyrosinase (29.9 × 10^3^%), NO (241.48%), MDA (742.86%), and COX-2 (288%), compared with the control group. Administration of PSOE after phenobarbital and formaldehyde injection (skin inflammation + PSOE group) significantly (*p* ≤ 0.05) reduced the levels of skin collagenase, elastase, hyaluronidase, tyrosinase, NO, MDA, and COX-2 by 80.0%, 70.0%, 72.5%, 83.3%, 65.3%, 57.6%, and 72.8%, respectively, compared with the administration of phenobarbital and formaldehyde only (Figure 2). In addition, the administration of Avocom-M after phenobarbital and formaldehyde injection (skin inflammation+ Avocom-M) significantly (*p* ≤ 0.05) decreased the levels of skin collagenase, elastase, hyaluronidase, tyrosinase, NO, MDA, and COX-2 by 90%, 80%, 95%, 90%, 72.34%, 13.22%, and 74.59%, respectively, compared with administration of phenobarbital and formaldehyde only (Figure 2). The rats treated with only PSOE for 21 days (PSOE group) showed no change in the levels of skin collagenase, elastase, hyaluronidase, tyrosinase, NO, MDA, or COX-2, compared with the control group (Figure 2).

Figure 3 compares the treatment effects of PSOE and Avocom-M in the rat study. The percentage decreases in the levels of skin collagenase, elastase, hyaluronidase, tyrosinase, NO, MDA, and COX-2 in the skin inflammation + PSOE group, compared with the skin inflammation group, were 80.0%, 0.0%, 72.5%, 83.3%, 65.3%, 57.6%, and 72.8%, respectively. The percentage decreases in the levels of skin collagenase, elastase, hyaluronidase, tyrosinase, NO, MDA, and COX-2 in the skin inflammation + Avocom-M group, compared with the skin inflammation group, were 90%, 80%, 95%, 90%, 72.34%, 13.22%, and 74.59%, respectively (Figure 3).

### 2.4. Patient Questionnaire Results

We determined reoccurring issues in the human subjects using a questionnaire, in which the subjects reported a score of 0% for hand calluses, 60% for nail inflammation, and 67% for extra skin tags around the nails after at least 2–3 months after treatment with PSOE. For TROSYD, the respondents reported a score of 50% for nail inflammation and 57% for extra skin tags around the nails after at least 2–3 months of treatment with TROSYD.

### 2.5. Biochemical Comparison of Three Groups in Human Investigations

Figure 4 shows a comparison of the three groups in the human study. Figure 4 shows the results for the different parameters studied, reported as mean ± SD for 20 volunteers in each group. Group 2 (PSOE treatment) showed significant declines in the levels of skin inflammation and skin collagenase, elastase, hyaluronidase, tyrosinase, NO, MDA, and COX-2: 78.6%, 78.8%, 66.6%, 80.9%, 47.4%, 55%, and 79%, respectively, compared with those in Group 1. Group 3 (TROSYD treatment) showed significant declines in the levels of skin collagenase, elastase, hyaluronidase, tyrosinase, NO, MDA, and COX-2 of 80%, 84.2%, 71.8%, 77.4%, 48.5%, 60.9%, and 79.8%, respectively, compared with those in Group 1 (Figure 4).

Figure 5 compares the treatment effects of PSOE and TROSYD in the human study. Group 2 (PSOE treatment), in comparison with Group 1, showed decreases in the levels of skin collagenase, elastase, hyaluronidase, tyrosinase, NO, MDA, and COX-2 of 78.6%, 78.8%, 66.6%, 80.9%, 47.4%, 55%, and 79%, respectively. Group 3 (TROSYD treatment), in comparison with Group 1, showed decreases of 80%, 84.2%, 71.8%, 77.4%, 48.5%, 60.9%, and 79.8% in these levels, respectively (Figure 5).

### 2.6. Biochemical Comparison of the Therapeutic Efficiency of PSOE between the Animal and Human Studies

Figure 6 compares the therapeutic efficiencies of PSOE between the animal and human studies. Comparing the skin inflammation + PSOE group with the skin inflammation group, the levels of skin collagenase, elastase, hyaluronidase, tyrosinase, NO, MDA, and COX-2 decreased by 80.0%, 70.0%, 72.5%, 83.3%, 65.3%, 57.6%, and 72.8%, respectively, in the rat study. Comparing Group 2 (PSOE treatment) to Group 1, the levels of skin collagenase, elastase, hyaluronidase, tyrosinase, NO, MDA, and COX-2 decreased by 78.6%, 78.8%, 66.6%, 80.9%, 47.4%, 55%, and 79%, respectively, in the human study (Figure 6).

Figure 7 depicts measurements of the reoccurrence of inflammation after stopping the therapy for 21 days in the human study. Figure 7 reports the levels of collagenase, elastase, hyaluronidase, tyrosinase, nitric oxide (NO), malondialdehyde (MDA), and cyclooxygenase2 (COX-2), where data are reported as mean ± SD, for 20 volunteers in each group. Figure 8 shows the recurrence of inflammation activity as a percentage change. Comparing Group 2 (PSOE treatment group) after treatment for 21 days with Group 2 after stopping treatment with PSOE for 21 days, the levels of skin collagenase, elastase, hyaluronidase, tyrosinase, NO, MDA, and COX-2 increased (by 132%, 16%, 15.5%, 177%, 5.7%, 18.7%, and 233%, respectively). Comparing Group 3 (TROSYD treatment) after treatment for 21 days with Group 3 (TROSYD treatment) after stopping treatment for 21 days, the levels of skin collagenase, elastase, hyaluronidase, tyrosinase, NO, MDA, and COX-2 increased by 79.3%, 0.0%, 15.8%, 119%, 1.79%, 3.5%, and 204%, respectively (Figure 8).

## 3. Discussion

Phenobarbital [27,28,29] and formaldehyde are present in our daily lives in chemicals, pharmacological treatments, and food additives; however, they are carcinogenic and can cause health issues, including inflammation, and neuropsychological disturbances, such as depression and stress [30,31,32,33,34]. Stress can be initiated in the body by chemicals and heavy metals [1,30], as well as physical, emotional, and psychiatric factors. Psychological stress occurs when individuals are under mental, physical, or emotional pressure that exceeds their ability to adapt to it or cope with it. As a consequence, the brain responds to the hormones secreted in response, such as dopamine, epinephrine, and cortisol, and triggers imbalances, which lead to physiological and behavioral disorders and changes, including skin disturbances [23,35].

Extended conditions of chronic stress, such as cancer treatment [30], may lead to hyperactivity of the hypothalamic–pituitary–adrenal (HPA) axis and increases in lipid peroxidation and inflammatory enzyme levels. In this study, the rat group with phenobarbital- and formaldehyde-induced neurological stress with skin inflammation showed elevated oxidative stress and inflammatory markers compared with the control group, in agreement with previously published results [36]. This situation may lead to maladaptive responses, which may increase pathological conditions, such as inflammatory skin disorders, autoimmune conditions, anxiety, and mood disturbances, especially in individuals with increased genetic susceptibility [23,37]. Psychological stress may initiate or worsen various skin diseases [35], which was supported by our findings. Our results in Figure 2 agreed with those of previous investigations wherein phenobarbital induced stress, was observed through skin inflammatory responses and lipid peroxidation. We found higher levels of collagenase, elastase, hyaluronidase, tyrosinase, nitric oxide (NO), malondialdehyde (MDA), and cyclooxygenase2 (COX-2) in the phenobarbital–formaldehyde group than in the control group in the rat study, and higher levels in Group 1 (control) than in Group 2 (PSOE treatment) volunteers in the human study (Figure 2 and Figure 4). Phenobarbital induces skin inflammation and enhances the incidence of skin carcinogenic processes [27]. This may be due to influencing hepatic-drug-metabolizing enzyme mechanisms and not necessarily through direct action on the skin [27,31]. Additionally, phenobarbital-related drugs have many neurological side effects that might also affect brain–skin conditions [28]. Formaldehyde is a toxic compound with many serious effects. Formaldehyde is present in many foods, drinks, cigarettes, and canned foods, and is an essential intermediary metabolite of various in vivo biochemical reactions [33]. Formaldehyde induces tumors, skin inflammation, and nasal adenoma in many species through direct interactions [33,34]. Formaldehyde directly connects with the mucus in the nasal epithelium, resulting in mucostasis and has direct effects on the epithelium, resulting in inflammation of the cilia. All body tissues may be affected through blood flow, thereby leading to many neurological, carcinogenic, and aging effects [33,34], which can manifest in neuroendocrinological disorders and the pro-oxidant initiation of skin inflammation problems [33,34].

In a recent study on the brain–skin connection, the skin was not only a target of modification through psychological stress signaling, but also strongly reflected the stress response through the local HPA axis, peripheral nerve terminations, and local skin cells, involving keratinocytes, mast cells, immune cells, and activated aging enzymes [35]. Additionally, treatments with certain drugs can induce stress, depression, and systemic inflammation, including skin inflammation [32,38]. Our findings were in agreement with this, showing increases in the levels of collagenase, elastase, hyaluronidase, tyrosinase, nitric oxide (NO), malondialdehyde (MDA), and cyclooxygenase2 (COX-2) in the phenobarbital–formaldehyde group, compared with the control group in the animal study, and in Group 1 (control) compared with Group 2 (PSOE treatment) human volunteers. Some bacteria, such as *Clostridium histolyticum*, use aging enzymes, known as bacterial collagenases, to strengthen their attack and prolong their infection of the skin [39]. From this perspective, our detected enzyme activity may have been due to the effects of phenobarbital and formaldehyde on the activity of any opportunistic bacteria [40] in both the rat and human studies. Our findings agreed with those of a previous study that indicated serious unfavorable cutaneous effects caused by phenobarbital [29].

The role of inflammatory enzymes and lipoperoxidation in skin inflammation, which leads to the development of skin disorders and aging, has been explained [2]. Skin aging, which includes photoaging, is one of the main skin issues. Photoaging and exposure to chemical compounds are external triggers of ROS synthesis, which results in the activation of collagenase, elastase, and hyaluronidase production in the cells and the overproduction of melanin, which is the rate-limiting enzyme. Tyrosinase also induces the inflammatory enzyme COX2 [2,30,31]. Additionally, the overproduction of ROS can cause lipid peroxidation, which is harmful to the DNA, initiates cell death, increases the chance of bacterial and fungal skin infections, and leads to many skin issues [41,42]. To initiate stress and mimic the neurological conditions that can trigger skin inflammation, aging, and increased chances of skin infection, we injected the rats in this study with an intramuscular dose of phenobarbital once per week for three weeks, plus a daily dose of intracutaneous formaldehyde [18,43]. Our methods conformed to the ethical standards of replacement, reduction, and refinement in scientific experiments [24]. The pharmacological treatment of skin issues mostly includes the selective application of creams, lotions, and alternative medicine interventions. Among these skin interventions are Avocom-M and TROSYD. Avocom-M cream, which contains 2% miconazole nitrate *w*/*w* and furoate 0.1% mometasone, is used as a treatment for neurological skin itching, severe allergic reactions, skin cancers, psoriasis treatment, and chronic skin infection, as well as an antifungal and antibacterial treatment [14,15]. TROSYD, containing 1% tioconazole, has anti-inflammatory and antifungal properties and is used to treat nail infections [15,44]. Our findings agreed with those of previous studies. We found that skin inflammation and itching, as well as the levels of collagenase, elastase, hyaluronidase, tyrosinase, nitric oxide (NO), malondialdehyde (MDA), and cyclooxygenase2 (COX-2), declined in rats after treatment with Avocom-M cream, compared with simultaneous treatment with phenobarbital and formaldehyde, and compared with the control group. This result from the rat study was confirmed by the results of the human investigation. We found significant decreases in the levels of collagenase, elastase, hyaluronidase, tyrosinase, nitric oxide (NO), malondialdehyde (MDA), and cyclooxygenase2 (COX-2), and decreases in nail inflammation and extra skin tags around the nails of the volunteers treated with TROSYD (Group 3), compared with those that did not receive the treatment (Group 1) (Figure 4). We also noted a minor improvement in the biological tests of rats that received Avocom-M treatment against phenobarbital–formaldehyde, compared with PSOE against phenobarbital–formaldehyde. We found a minor ameliorating effect in the humans treated with TROSYD (Group 3) compared with those treated with PSOE (Group 2), which might have been due to the concentrations of the chemicals in the pharmacological creams. As minor side effects are generally observed with pharmacological treatments, the use of natural cosmetic constituents from plants has become a target for studies [17,45,46,47].

In a study of papyrus in Egypt, the importance of the medical effects of the pomegranate tree and its fruits, including in wound healing, was mentioned in a previous publication [48]. Interestingly, a similar effect of the study was already noted in present outcomes. We report here that pomegranate seed oil extract treatment reversed the effects of phenobarbital and formaldehyde, suggesting that its extract might have anti-inflammatory, antifungal, and antibacterial effects. The results showed an improvement in the rats’ skin levels of collagenase, elastase, hyaluronidase, tyrosinase, nitric oxide (NO), malondialdehyde (MDA), and cyclooxygenase2 (COX-2) following PSOE treatment against phenobarbital and formaldehyde, with no effects in the group treated with PSOE alone. This agreed with our previous finding, where we observed no side effects on the biological profiles, including the liver and kidney profiles, DNA fragmentation, and histopathological markers at doses under 250 mL/kg following short- and long-term treatment with pomegranate seed oil extract [18,20,21,22]. Additionally, the results showed that the biochemical parameters in the humans were significantly ameliorated with PSOE treatment, compared with those in Group 1. In the current study, the beneficial effect of PSOE against skin inflammation and the modulation of the biochemical enzymes under study might have been due to a direct effect on the skin or through reversals of the effects of phenobarbital and formaldehyde through a detoxification process and systematic balancing of their toxicity. Furthermore, a synergetic effect of the phytochemical and polyphenol contents in pomegranate was reported in the treatment of asthma, allergies, inhalation of toxic chemicals, and inflammation [49,50,51,52].

Hence, previous findings [49,50,51,52] indicated that pomegranate phytochemicals had beneficial effects, in agreement with our findings. We report that PSOE includes necessary metals, such as potassium (K,) sodium (Na), calcium (Ca), iron (Fe), and zinc (Zn), and does not contain toxic heavy metals, such as lead (Pb), cadmium (Cd), or aluminum (Al), indicating that pomegranate extract is a healthy skin treatment (Table 1). In our previous study, we reported that K, Na, Ca, Fe, and Zn produced many beneficial health effects by decreasing C-reactive protein levels, as well as having anti-inflammatory and antioxidant effects. Additionally, these elements support both the immune and endocrine systems against opportunistic bacteria and fungi [1,51,53]. In both our past work and current studies, we found that pomegranate contains phytochemicals, including squalene, gamma-tocopherol, cis-11-eicosenamide, hexadecanamide, hydroxymethylfurfural, stigmastan-3,5-diene, oleamide, pentanoic acid, and octadecanoic acid, which have antiapoptotic, anti-inflammatory, antioxidant, antibacterial, and antifungal activities (Table 2) [22,23,54]. Additionally, our previous findings indicated that the active phytochemicals in pomegranate could be used in schizophrenia and depression treatment, as a sedative agent, and to treat neurological stress through the downregulation of epinephrine, cortisol, and inflammatory cytokine receptors, in agreement with our current findings [22,23,54].

In agreement with the outcomes reported here, a previous report indicated that PSOE contained a combination of many phytochemicals, including flavanones, isoflavones, flavonols, flavones, phenolic acids, anthocyanidins, minerals, vitamin precursors, proanthocyanidins, phytosterol, tannins, and hydroxy-benzoic acids [19,48]. These phytochemicals have immunomodulatory, antioxidant, antimicrobial, antifungal, anti-inflammatory [53], antibiofilm, and anti-quorum-sensing effects in the wound healing process, and, moreover, pomegranate extracts increase the action of tissue-repair mechanisms [55,56,57]. Furthermore, Ramírez-Boscá et al. (2015) recommended the application of pomegranate seed oil as a cosmetic component and anti-aging agent [55,56,57]. Biological inhibition of the activities of aging enzymes tyrosinase, elastase, collagenase, and hyaluronidase were confirmed in the current study and could be helpful for application in cosmeceutical aging treatments, in agreement with a previous skin treatment investigation that reported that pomegranate seed oil helped the regeneration of epidermis [58]. The previous findings approximated the results of the current work, which found beneficial effects of PSOE, Avocom-M, and TROSYD, which acted to diminish the level of ROS, such as MDA and NO, leading to decreased levels of inflammatory enzyme COX2, and, consequently, modulating skin inflammation problems. As a result, PSOE, Avocom-M, and TROSYD helped to restore the balance of skin dermis enzymes, such as collagenase, elastase, hyaluronidase, and tyrosinase significantly, compared to controls in both animal and human observational studies.

Our findings also demonstrated that the reoccurrence of skin issues was 0% for hand calluses, 60% for nail inflammation, and 67% for extra skin tags around the nails after at least 2–3 months of the treatment with PSOE. For TROSYD, the reoccurrence was 50% for nail inflammation and 57% for extra skin tags around the nails after at least 2–3 months of treatment (Figure 7 and Figure 8). The recurrence of inflammation in the human study after stopping the therapy for 21 days may have been due to dose and treatment duration. As such, further long-term studies with a larger sample, different doses, different types of extract, and both sexes are needed to confirm and validate the results.

Overall, we noted an improvement the rats’ skin condition with the use of PSOE and Avocom-M treatment against the effects of phenobarbital and formaldehyde. We observed significantly improved human skin reports following experimental PSOE and TROSYD treatment. PSOE treatment had beneficial effects on the skin levels of collagenase, elastase, hyaluronidase, tyrosinase, nitric oxide (NO), malondialdehyde (MDA), and cyclooxygenase2 (COX-2) in humans compared with rats. This may have been due to the phytochemical content of the pomegranate seed oil, as well as the chemical compositions of the Avocom-M and TROSYD creams.

## 4. Materials and Methods

### 4.1. Plant Materials

We obtained pomegranates (*Punica granatum*) from a local market in Saudi Arabia for this study. Figure 1 shows the experimental design. We crushed and extracted the pomegranate with a solvent (1: 4 *w*/*v*) composed of hexane and ethanol (3:1). After soaking in the solvent for 20 min at room temperature (26 ± 3 °C), we separately concentrated the extracts in a 40–50 °C rotary evaporator, according to our previously reported extraction technique [22]. We obtained a total PSOE yield of 30% oil.

#### 4.1.1. Metal Analyses

We applied our previous process for inductively coupled plasma–mass spectrometry (ICP–MS; 7500 cx, Agilent Technologies, Santa Clara, CA 95051, USA) [1]. After digesting the PSOE with nitric acid (1:1 *v*/*v*) in a microwave digestion system Ethos 1 (Milestone, Fremont, CA 94539, USA), according to the manufacturer’s instructions [1], we diluted the PSOE sample (1:4 *v*/*v*) with nitric acid to examine the concentrations of potassium (K,) sodium (N.), calcium (Ca), iron (Fe), zinc (Zn), lead (Pb), cadmium (Cd), and aluminum (Al) in PSOE [1].

#### 4.1.2. Gas Chromatography–Mass Spectrometry Analysis

We followed our previously reported method for this GC–MS investigation to identify the phytochemical contents of the PSOE using a gas chromatography system (G3440B, Agilent Technologies, Santa Clara, CA 95051, USA). We redissolved the PSOE in ethanol:hexane (1:4), which we then distilled through a nylon 0.45 μm pore-size membrane filter, according to the manufacturer’s instructions [22]. We injected 2 µL of the sample into the GC-MS system, and we ran the test as a general screening analysis using helium as a carrier with a 1 mL/min flow rate, according to the manufacturer’s instructions [22]. We then used the WILEY and National Institute of Standards and Technology (NIST) mass spectral libraries to identify the chemicals included in the PSOE extract, as described in our previous study [22].

### 4.2. Chemicals

We purchased the COX-2 activity assay (catalog kit no.760151) from Cayman (Coral Springs, FL 33076, USA), and the collagenase activity assay (MAK293) from Sigma-Aldrich, (St. Louis, MO, USA). We purchased the elastase activity assay (Colorimetric Drug Discovery Kit BML-AK497) from Enzo Life Sciences, Inc. (Farmingdale, NY 11735, USA). We purchased the hyaluronidase and tyrosinase activity assays (Catalog # E4909-100; catalog # K742-100), respectively, from Biovision (Milpitas, CA 95035, USA). We purchased N-(1-naphthyl) ethylenediamine, sulfanilamide, standard sodium nitrite, sodium dodecyl sulfate, thiobarbituric acid, tetramethoxypropane, phenobarbital, and formaldehyde from Sigma-Aldrich, (St. Louis, MO, USA). We obtained TROSYD, 1% Tioconazole Dermal Cream 50 g Pfizer (235 East 42nd Street in the Turtle Bay neighborhood, NY 10017, USA) and Avocom-M Cream (50 g, product code: 010530013) from a local pharmacy.

### 4.3. Animal Study

We used 40 adult male Sprague–Dawley rats, each weighing 120–150 g, in this study. We tested the health status of these rats at 28 °C, and we fed them a specified standard diet daily and water ablibitum for one week before the study started. After adaptation, we separated the rats into five groups of eight rats each. We conducted all animal experiments according to the Experimental Animal Care Society’s Department Committee guideline, and adapted to the three Rs (replacement, reduction, and refinement) [23,24,25,26]. The five groups of rats were as follows: a control group of untreated rats; a skin inflammation group, in which rats were intramuscularly injected with a dose of phenobarbital at 50 mg/kg body mass (bm) once per week for three weeks plus a daily intracutaneous dose of formaldehyde at 6 µL/kg body mass (bm) [18,43,59]. In the PSOE group, the rats were orally administered a daily dose of PSOE (250 mL/kg bm) for 21 days [18]. In the skin inflammation + PSOE group, rats were injected with phenobarbital, formaldehyde, and PSOE at the same dosage and in the same manner for three weeks; additionally, PSOE (250 mg/kg) treatment was administered directly to the inflamed part of the skin once per day for 21 days. In the skin inflammation + Avocom-M group, rats were injected with phenobarbital and formaldehyde at the same dosage and in the same manner as the phenobarbital and formaldehyde groups for three weeks; additionally, Avocom-M 250 mg/kg treatment was administered directly to the inflamed part of the skin once per day for 21 days.

Next, once the treatment period finished, we withdrew food from the rats for 12 h before they were anesthetized with diethyl ether and sacrificed. We obtained a skin tissue sample from each rat and washed it with a cold saline solution (0.9% NaCl), weighed it, and kept it at −80 °C until the biochemical investigation.

### 4.4. Human Study

#### Patient Population and Data Collection

This investigation included 60 women with skin issues, including hand calluses, nail inflammation, and extra skin tags around the nails. Patients with a history of any other skin disease were excluded from the study. Obese individuals, pregnant women, smokers, and children were also excluded. The selected volunteers were 25–66 years of age, women, and had a BMI of 33.58 ± 2.1. The appropriate institution below approved our protocol, which fulfilled the requirements of the Declaration of Helsinki as reviewed in 2013. Written informed consent was obtained from all the volunteers [1,23]. We divided the volunteers into three groups—Group 1 was not given any skin treatment (for at least 2–3 months), Group 2 was administered PSOE as a skin treatment, and Group 3 was administered TROSYD as prescribed by a dermatologist (for 21 days), as in our previous study [1,23,37,60,61].

We collected skin swab samples from areas of nail inflammation, tiny parts from extra skin tags around the nails, and the data for investigation of parameters in this observational study after each volunteer had received 21 days of treatment with PSOE (250 mL/kg) and TROSYD 250 mg/kg. The treatments were administered directly to the inflamed parts of the skin once a day for 21 days of treatment. We also collected skin swab samples and the data for investigation of parameters in this observational study 21 days after each volunteer had stopped the treatment with PSOE and TROSYD to investigate the recurrence of the skin issues. We also assessed the volunteers’ recurring skin issues using a questionnaire [1,23,37,53,60,61].

### 4.5. Biochemical Assays

#### 4.5.1. Collagenase

We assayed collagenase enzyme activity in the animal skin and human skin swab samples separately, according to the manufacturer’s instructions. We measured the collagenase levels in samples using a colorimetric method MAK293, Sigma-Aldrich, (St. Louis, MO, USA). Human skin swab and rat skin samples were separately homogenized in (1:4) volumes of cold water using a Teflon glass homogenizer. The homogenates were centrifuged at 20,000× *g* for 20 min at 4 °C [62], and the supernatants were kept at −80 °C. We placed 10 μL of each sample in a microplate reader, and then 90 μL reaction buffer was added. Hence, for each 100 μL of reaction buffer, we mixed 40 μL of collagenase substrate (the peptidic collagenase substrate N-[3-(2-furylacryloyl)]-l-leucyl-glycyl-l-prolyl-l-alanine (FALGPA) and 60 μL of collagenase assay buffer, according to the manufacturer’s instructions [56]. Optical densities were reported using an absorbance microplate reader (Bio-Tek Instruments, Bad Friedrichshall, Germany) at 345 nm in a microplate reader at 37 °C for 5–15 min (kinetic mode) against a blank. Collagenase activity was determined in U/mL according to the kit protocol equation [56].

#### 4.5.2. Elastase

We assayed elastase enzyme activity in the animal skin and human skin swab samples separately, according to the manufacturer’s instructions. We measured the elastase levels in the samples using a colorimetric method colorimetric drug discovery kit BML-AK497, Enzo Life Sciences, Inc. (Farmingdale, NY 11735, USA). Human swabs and rat samples were separately homogenized in 1:10 volumes of cold assay buffer (including 100 mM HEPES, 500 mM NaCl, 0.05% Tween-20, pH 7.25) using a Teflon glass homogenizer. The homogenates were centrifuged at 13,000× *g* for 20 min at 4 °C [63,64], and the supernatants were kept at −80 °C. We placed 10 μL of each sample in a microplate reader, and then 5 μL substrate (the chromogenic substrate MeOSucAla-Ala-Pro-Val-pNA) and 65 μL reaction buffer was added, according to the kit manufacturer’s instructions [48]. Optical densities were reported using an absorbance microplate reader (Bio-Tek Instruments, Bad Friedrichshall, Germany) at 405 nm in a microplate reader against a blank at 37 °C. Data were recorded at 1 min time intervals for 5–10 min. We then applied different concentrations of calibration standard using the assay buffer as a blank, according to the manufacturer’s instructions [65]. Elastase activity was determined according to the kit protocol equation in pmol/min [66].

#### 4.5.3. Hyaluronidase

We separately assayed hyaluronidase enzyme (HYAL) activity in the rat skin and human skin swab samples, according to the manufacturer’s instructions. We measured the hyaluronidase level in each sample using Biovision’s (Milpitas, CA 95035, USA) hyaluronidase activity ELISA kit (Catalog # E4909-100). We separately homogenized human swab and rat samples in 1:4 volumes of cold HYAL assay buffer with a Teflon glass homogenizer. We centrifuged the homogenates at 10,000× *g* for 20 min at 4 °C. Then, we mixed 0.2 mL of the supernatant with 0.2 mL of HYAL assay buffer. We kept the supernatants at −80 °C. We placed 100 μL of each sample in a precoated biotin–hyaluronic acid (HA) 96-well plate. We used a sequential 5-fold serial dilution using 4 U/mL HYAL standard solution to prepare the standard curve, and we added 100 μL of each prepared standard solution into the appropriate wells. We washed the wells five times with 250 μL of wash buffer for 30 s each time. We added 50 μL of working buffer to each well according to the manufacturer’s instructions. Then, we added 100 μL of substrate to each well. We recorded optical densities using absorbance at 450 nm at room temperature within 15 min of adding 50 μL of stop solution. We determined hyaluronidase activity, according to the kit’s standard curve protocol. The specific enzyme activity was calculated in mU/g [56,66,67].

#### 4.5.4. Tyrosinase

We separately assayed tyrosinase enzyme activity in the animal skin and human skin swab samples, according to the manufacturer’s instructions. We measured the tyrosinase level in a sample using a Biovision (Milpitas, CA 95035, USA) tyrosinase activity assay colorimetric kit (Catalog # K742-100). We separately homogenized human swab and rat samples in 1:10 volumes of cold assay buffer with a Teflon glass homogenizer, according to the kit protocol. We centrifuged the homogenates at 10,000× *g* for 15 min at 4 °C, and we kept the supernatant at −80 °C. We placed 25 μL of the supernatant of each sample and 25 μL of substrate background (as a background control) in a microplate reader; we adjusted the volume in each well to 50 µL with tyrosinase assay buffer. We obtained the chromophore standard curve according to the kit protocol with a series of wells in a clear 96-well plate. We used tyrosinase assay buffer to adjust each well to 100 μL, which we mixed at 37 °C. Then, we prepared the reaction buffer according to the kit protocol, and mixed 50 μL of reaction buffer in each well immediately before recording the absorbance for 30 s.

We measured optical densities using an absorbance microplate reader (Bio-Tek Instruments, Bad Friedrichshall, Germany) at 510 nm. We immediately started recording absorbance in 30 s intervals for 10–15 min. In this assay, tyrosinase catalyzed the process of a phenolic substrate to a quinone intermediate, which interacted with the tyrosine enhancer, thereby producing a stable chromophore with absorbance at 510 nm. We determined tyrosinase activity, according to the kit’s protocol standard curve in endpoint mode. We calculated the specific enzyme activity (mU/μg) following the protocol equation provided with the kit [56,66,67].

#### 4.5.5. NO Level

We separately analyzed the NO levels in the animal skin and human skin swab samples via a spectrophotometry assay, according to Montgomery and Dymock [68]. A comprehensive account of the assay was provided in our previous study [54].

#### 4.5.6. Lipid Peroxidation

We separately analyzed the levels of malondialdehyde (MDA), the end product of lipid peroxidation, in the rat skin and human skin swab samples, using a colorimetric method, according to Ohkawa et al. [54,69]. Our previous study provided the details of this examination [54].

#### 4.5.7. COX-2

We separately analyzed the COX-2 enzyme levels in the rat skin and human skin swab samples, according to Smith et al. [23,54,70]. The particulars of this analysis were provided in our prior study [23].

### 4.6. Statistical Analyses

We analyzed the data using IBM SPSS software version 20.0 (IBM Corp, Armonk, NY, USA). We set the significance level of the tests to 5% [23].

## 5. Conclusions

We revealed the mitigation of the effects of phenobarbital- and formaldehyde-induced neurological oxidative stress and skin inflammation by Avocom-M and PSOE. Treatment with either of these extracts decreased the levels of collagenase, elastase, hyaluronidase, tyrosinase, cyclooxygenase-2, lipid peroxidation, and nitric oxide in rats, compared with the levels observed in the rats treated with phenobarbital and formaldehyde. The outcomes of the rat study were confirmed by comparisons of the data from human volunteers treated with TROSYD or PSOE with those from untreated volunteers. We concluded that PSOE can be used as a local treatment for skin issues. Further studies are underway.

## Figures and Tables

**Figure 1 molecules-28-00903-f001:**
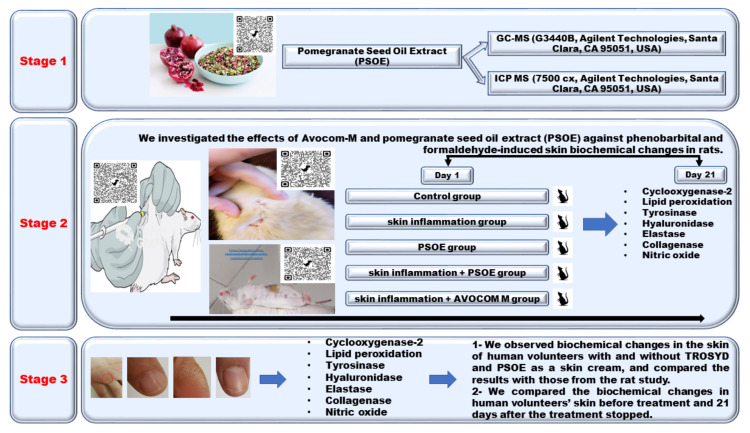
Experimental design [23,24,25,26].

**Figure 2 molecules-28-00903-f002:**
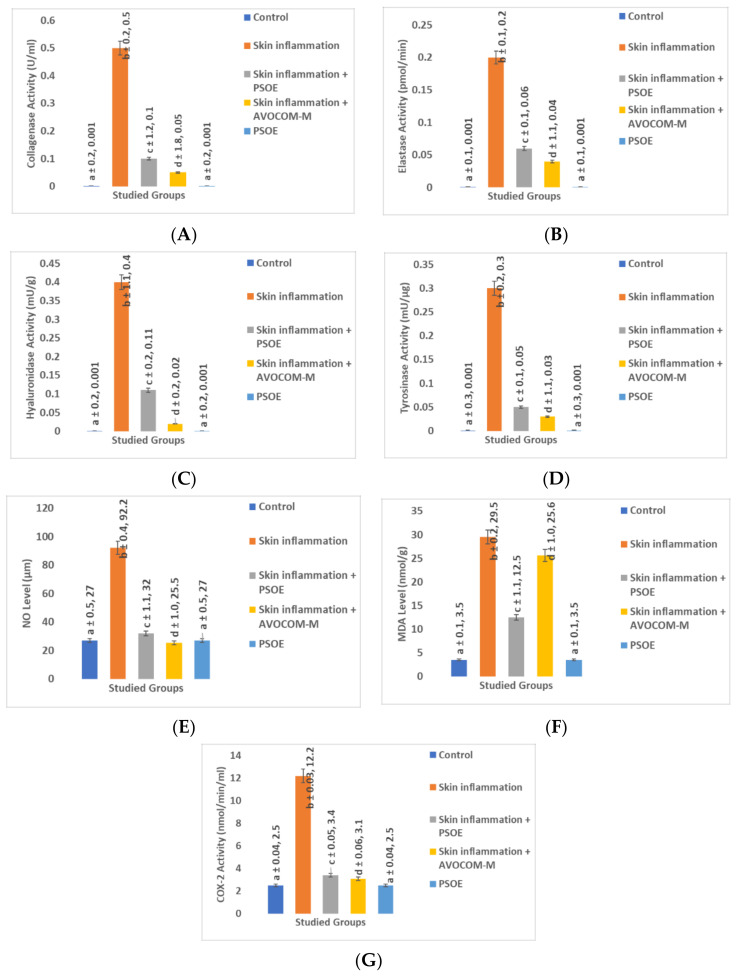
Biochemical comparison of different groups in rat skin investigation. (**A**) Collagenase; (**B**) elastase; (**C**) hyaluronidase; (**D**) tyrosinase; (**E**) nitric oxide (NO); (**F**) malondialdehyde (MDA); and (**G**) cyclooxygenase2 (COX-2) levels in five rat groups. Rats in the control group (**C**) were untreated; rats in the skin inflammation group were injected with an intramuscular dose of phenobarbital at 50 mg/kg body mass (bm) once per week for three weeks, plus a daily dose of intracutaneous formaldehyde at 6 µL/kg body mass (bm); rats in the PSOE group received a daily oral dose of PSOE at 250 mL/kg bm for 21 days; rats in the skin inflammation + PSOE group were injected with phenobarbital, formaldehyde, and PSOE at the same dosages and in the same manner as the phenobarbital, formaldehyde, and PSOE groups for three weeks; additionally, 250 mg/kg PSOE treatment was administered directly to the inflamed skin once per day for 21 days of treatment; rats in the skin inflammation + Avocom-M group were injected with phenobarbital and formaldehyde at the same dosages and in the same manner as the phenobarbital and formaldehyde groups for three weeks; we also administered Avocom-M 250 mg/kg treatment directly to the inflamed skin once per day for 21 days of treatment. Data are given as mean ± SD for 8 rats. Statistical significance was set to *p* ≤ 0.05; means reported with the same letters (a, b, c, d) were not significantly different.

**Figure 3 molecules-28-00903-f003:**
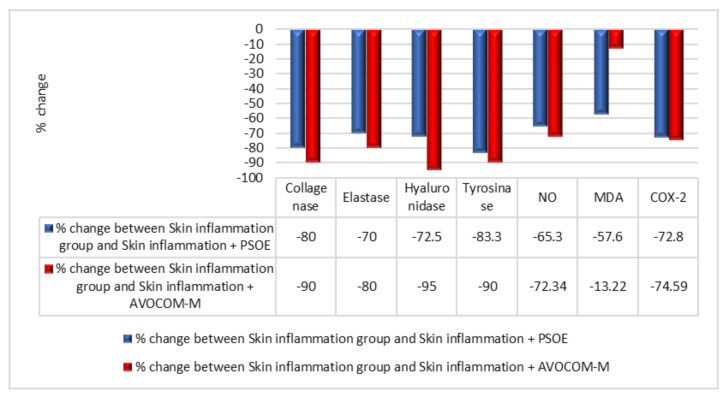
Comparison of beneficial effects of pomegranate seed oil extract (PSOE) and Avocom-M treatment in animal skin investigations. Collagenase, elastase, hyaluronidase, tyrosinase, nitric oxide (NO), malondialdehyde (MDA), and cyclooxygenase2 (COX-2) levels in five rat groups were compared in terms of percentage change between the skin inflammation + PSOE group and skin inflammation group, and between the skin inflammation + Avocom-M and skin inflammation groups.

**Figure 4 molecules-28-00903-f004:**
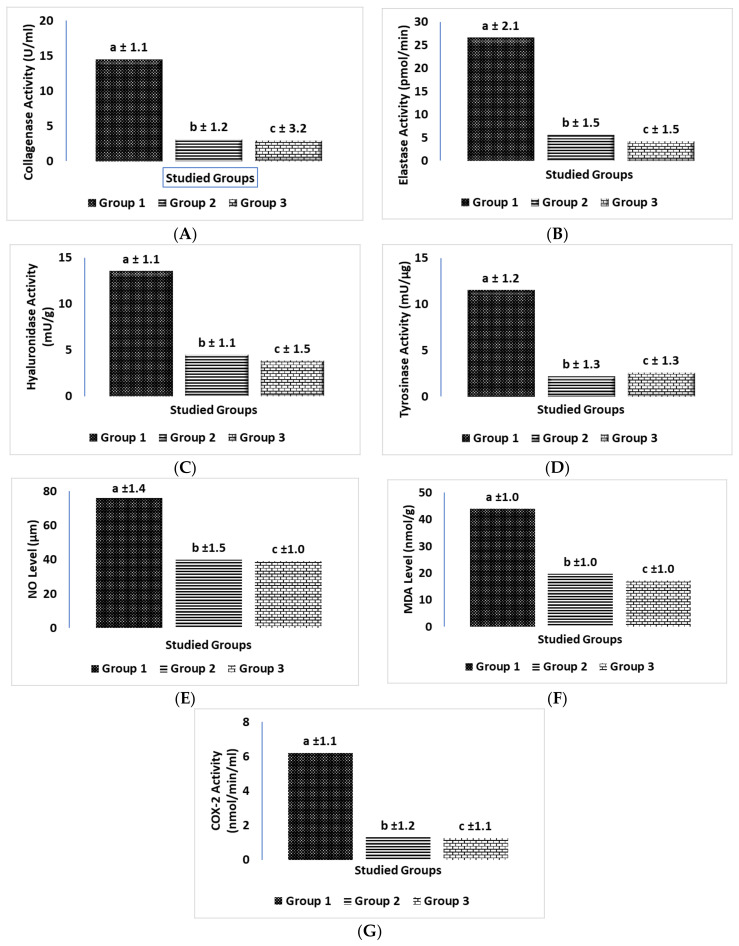
Biochemical comparison of the three groups in human skin investigation. (**A**) Collagenase; (**B**) elastase; (**C**) hyaluronidase; (**D**) tyrosinase; (**E**) nitric oxide (NO); (**F**) malondialdehyde (MDA); and (**G**) cyclooxygenase2 (COX-2) levels were compared between Group 1, who did not receive any skin treatment (for at least 2–3 months); Group 2 members, who were administered 250 mL/kg PSOE treatment; and Group 3 members, who were administered 250 mg/kg TROSYD for 21 days. Treatments were administered directly to the inflamed skin once per day for 21 days of treatment. Data are shown as mean ± SD for 20 volunteers in each group. Statistical significance was set at *p* ≤ 0.05; means reported with the same letters (a, b, c) were not significantly different.

**Figure 5 molecules-28-00903-f005:**
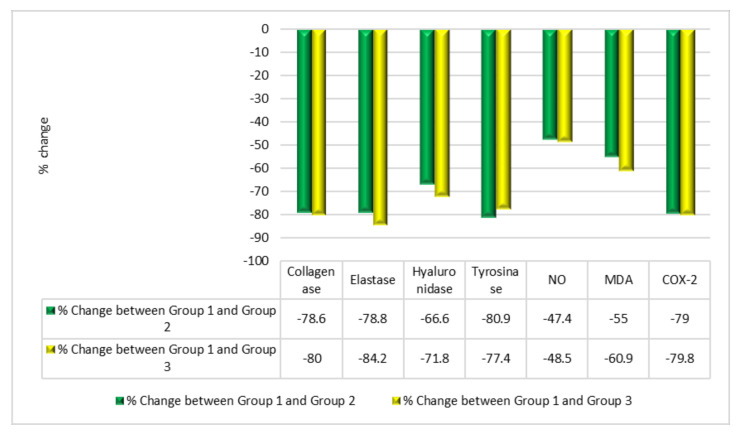
Comparison of beneficial effects of pomegranate seed oil extract (PSOE) and TROSYD treatment in human skin investigations. We compared the changes in the levels of collagenase, elastase, hyaluronidase, tyrosinase, nitric oxide (NO), malondialdehyde (MDA), and cyclooxygenase 2 (COX-2) between Group 1 and Group 2 to the changes between Group 1 and Group 3. Group 1 received no skin treatment (for at least 2–3 months); Group 2 was administered 250 mL/kg PSOE treatment; Group 3 was administered 250 mg/kg TROSYD for 21 days by a dermatologist. We administered the treatments directly to the inflamed part of the skin once per day for 21 days of treatment.

**Figure 6 molecules-28-00903-f006:**
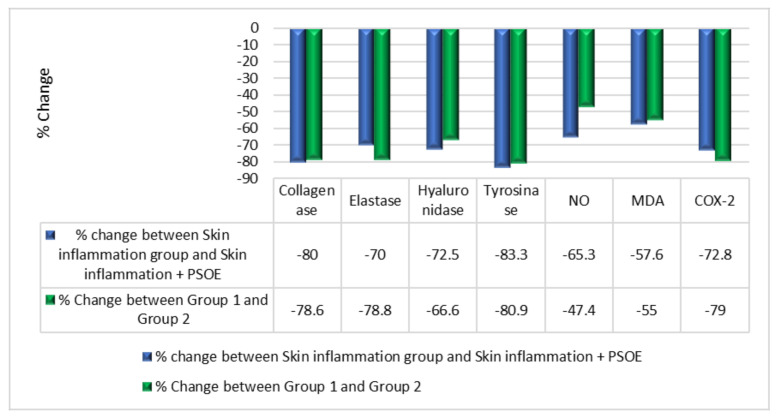
Comparison of beneficial effects of pomegranate seed oil extract (PSOE) between rat and human investigations. We compared the levels of collagenase, elastase, hyaluronidase, tyrosinase, nitric oxide (NO), malondialdehyde (MDA), and cyclooxygenase2 (COX-2) between the skin inflammation + PSOE group and the skin inflammation group (Group 1 and Group 2). We intramuscularly injected the rats in the skin inflammation group with a dose of phenobarbital (50 mg/kg body mass (bm)) once per week for three weeks, plus a daily dose of intracutaneous formaldehyde at 6 µL/kg of body mass (bm). We orally administered rats in the PSOE group a daily dose of PSOE (250 mL/kg bm) for 21 days. The rats in the skin inflammation + PSOE group were injected with phenobarbital, formaldehyde, and PSOE at the same dosages and in the same manner for three weeks, plus 250 mg/kg PSOE was administered directly to the inflamed part of the skin once per day for 21 days. Additionally, Group 1 did not receive any skin treatment (for at least 2–3 months); Group 2 was administered PSOE (250 mL/kg) as a skin treatment. In the human study, the treatments were administered directly to the inflamed part of the skin once per day for 21 days.

**Figure 7 molecules-28-00903-f007:**
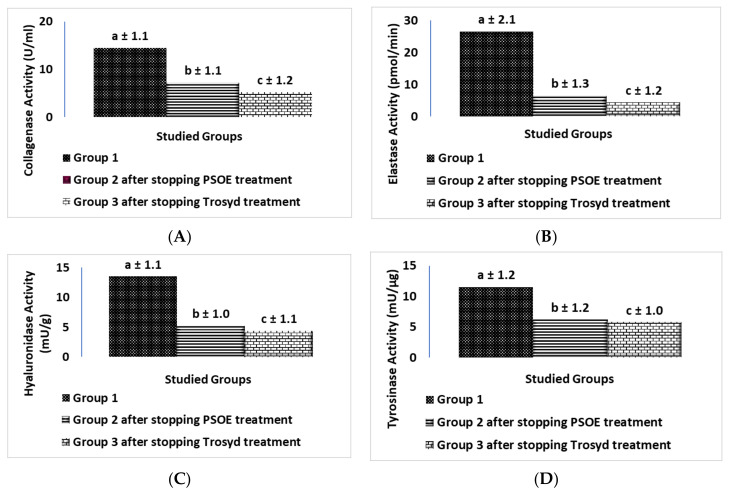
Reoccurrence of inflammation after stopping therapy for 21 days in the human study. (**A**) Collagenase, (**B**) elastase, (**C**) hyaluronidase, (**D**) tyrosinase, (**E**) nitric oxide (NO), (**F**) malondialdehyde (MDA), (**G**) cyclooxygenase2 (COX-2) levels; Group 1, who did not receive any skin treatment (for at least 2–3 months); Group 2, (PSOE treatment group) after stopping treatment for 21 days; Group 3, (TROSYD treatment) after stopping treatment for 21 days. Data are shown as mean ± SD for 20 volunteers in each group. Statistical significance was detected at *p* ≤ 0.05; means reported with the same letters (a, b, c) were not significantly different.

**Figure 8 molecules-28-00903-f008:**
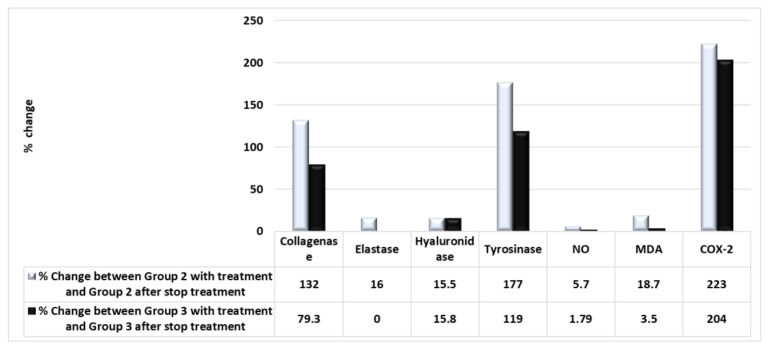
The recurrence of inflammatory activity as a percentage change. Comparison between Group 2 (PSOE treatment group) with treatment for 21 days with Group 2 after stopping treatment with PSOE after 21 days, showing recurrence of inflammatory activity as a percentage change. Comparison between Group 3 (TROSYD treatment) with treatment for 21 days with Group 3 after stopping treatment with TROSYD treatment after 21 days, showing recurrence of inflammatory activity as a percentage change.

**Table 1 molecules-28-00903-t001:** Analysis of metal contents in PSOE.

Metal	Concentration (mg/kg)
Na	2.92
K	220
Ca^2+^	8.60
Fe	0.26
Zn	0.39
Cd	Not detected
Pb	Not detected
Al	Not detected

**Table 2 molecules-28-00903-t002:** Gas chromatography–mass spectrometry analysis of PSOE (pomegranate (*Punica granatum*) seed oil extract).

Sr. No.	Name	Base Peak	RT *	Chromatogram (Device Print Out)
1.	5-Hydroxymethylfurfural	97	5.702–6.087	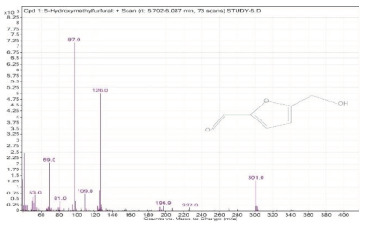
2.	Pentanoic acid, 5-hydroxy-, 2,4-di-t-butylphenyl esters	191.2	8.227–8.291	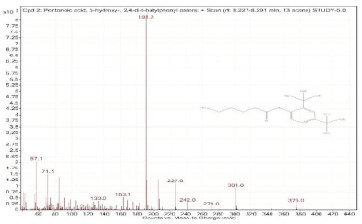
3.	Dodecyl acrylate	55	9.554–9.591	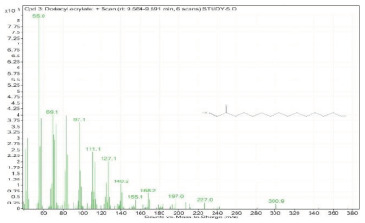
4.	n-Hexadecanoic acid	73	11.527–11.628	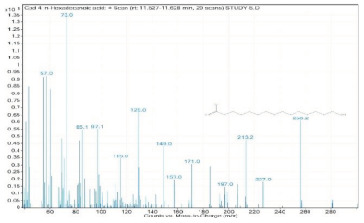
5.	Oleanitrile	55	12.302–12.351	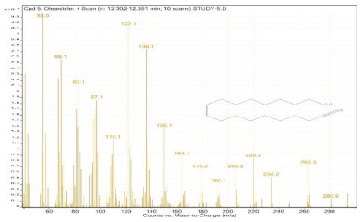
6.	9-Octadecenoic acid, methyl ester, (E)-	55.1	12.351–12.383	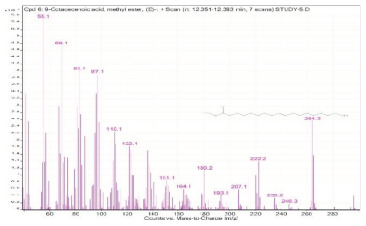
7.	9,12-Octadecadienoic acid (Z,Z)-	55	12.575–12.778	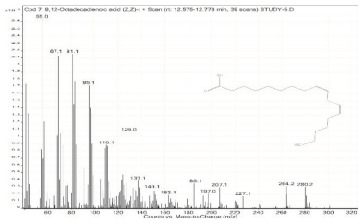
8.	Octadecanoic acid	55	12.778–12.880	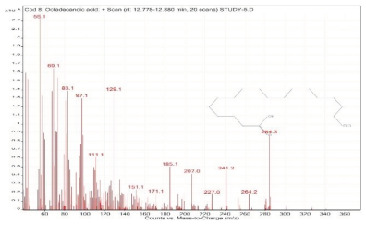
9.	Hexadecanamide	59	12.934–12.960	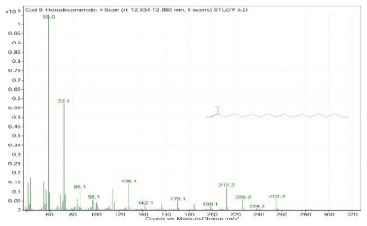
10.	7-Hexadecenal, (Z)-	55	13.522–13.624	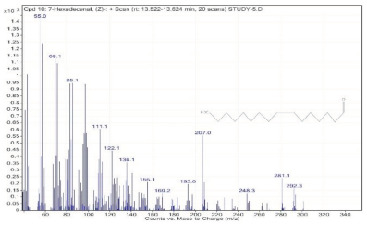
11.	9-Octadecenamide, (Z)-	59	14.003–14.073	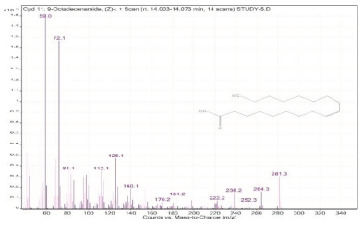
12.	9-Octadecenamide, (Z)-	59	14.073–14.126	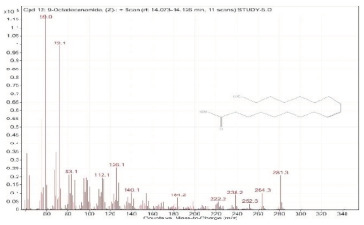
13.	Octadecanamide	59	14.126–14.180	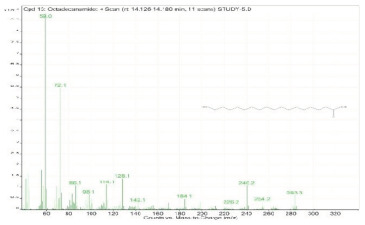
14.	Oleamide	59	14.239–14.340	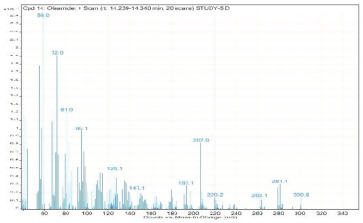
15.	Erucic acid	55.1	14.645–14.677	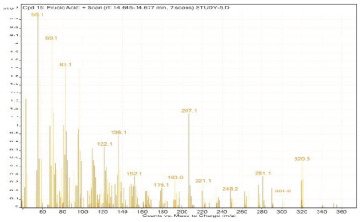
16.	cis-11-Eicosenamide	59	15.127–15.164	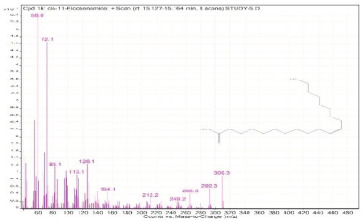
17.	cis-11-Eicosenamide	59	15.164–15.223	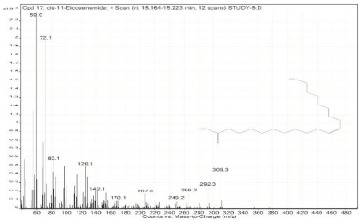
18.	9-Octadecenamide, (Z)-	59	15.233–15.292	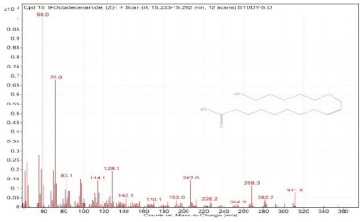
19.	13-Docosenamide, (Z)-	59	16.458–16.506	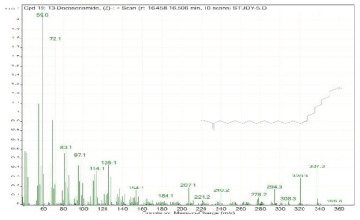
20.	Squalene	69.1	16.517–16.565	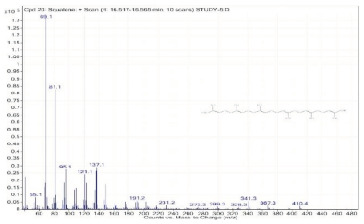
21.	Stigmastan-3,5-diene	145.1	18.213–18.277	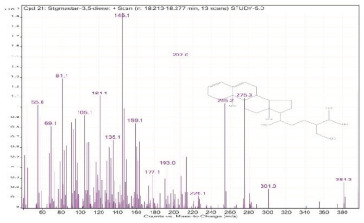
22.	gamma-Tocopherol	416.4	19.015–19.101	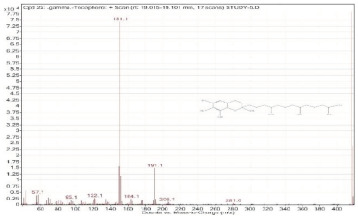
23.	Stigmastan-3,5-diene	396.4	19.785–19.865	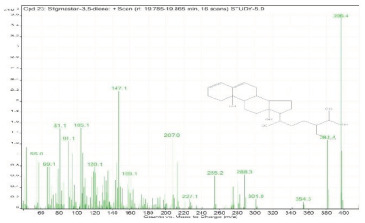

* RT: retention time (minutes); PA: peak area (%). GC-MS investigation was performed for hexane:ethanoic (3:1) extracts with a G3440B (Agilent Technologies, Santa Clara, CA 95051, USA). PSOE contents were investigated through computer simulations using the Wiley and National Institute of Standards and Technology (NIST) commercial libraries.

## Data Availability

No new data were created or analyzed in this study. Data sharing is not applicable to this article.

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
