# Peer review of "Biochemical Pilot Study on Effects of Pomegranate Seed Oil Extract and Cosmetic Cream on Neurologically Mediated Skin Inflammation in Animals and Humans: A Comparative Observational Study"

_molecules, 2023, doi:10.3390/molecules28020903_

Round 1
Reviewer 1 Report
Key words should appear in alphabetic order
Graphics are low resolution.
Author Response
- Dear reviewer with due all respect, it is our honor to received your revision to our work, that gives it more valuable and clarity.
Point 1: Key words should appear in alphabetic order .
Response 1: Done.
Point 2: Graphics are low resolution.
Response 2: Done.
Explanation:
- The graphs inside the table are printouts directly from the device, and we increased the resolution to 300 dpi according to journal policy.
- All photos are 300 dpi but if the size of the photos changed during the revision process the dpi resolution lowers with it, we put it again as 300 dpi, and if it changed again after editing we will put it in the final copy.

Reviewer 2 Report
The topic should be of interest to the readers.
Although the manuscript is technically sound for the most part, it requires revisions.
Specifically, although the manuscript investigate neurological mediated skin inflammation, the authors failed to even mention that the skin has powerful neuroendocrine properties, which are already recognized for more than 20 years ((Endocrine Rev 21, 457-487, 2000). The most recent advances on this topic in the epidermis including neuropeptides have been discussed recently (American Journal of Physiology-Cell Physiology 2022 323:6, C1757-C1776). Both have to be properly mentioned already in the introduction.
The readers would also appreciate short information on different mechanism regulating melanin pigmentation (Physiol Rev 84, 1155-1228, 2004; Pigment Cell Melanoma Res 25, 14-27, 2012) and mentioning various properties of melanin under pathological and physiological conditions (Frontiers in Oncology 2022;12. DOI: 10.3389/fonc.2022.842496).
Finally, the readers would appreciate histological figures with the description.
Author Response
- Dear reviewer with due all respect, it is our honor to received your revision to our work, that gives it more valuable and clarity.
Point 1: The topic should be of interest to the readers.
Response 1: Done
Point 2: Although the manuscript is technically sound for the most part, it requires revisions.
Response 2: Done
Point 3: Specifically, although the manuscript investigate neurological mediated skin inflammation, the authors failed to even mention that the skin has powerful neuroendocrine properties, which are already recognized for more than 20 years ((Endocrine Rev 21, 457-487, 2000). The most recent advances on this topic in the epidermis including neuropeptides have been discussed recently (American Journal of Physiology-Cell Physiology 2022 323:6, C1757-C1776). Both have to be properly mentioned already in the introduction.
Response 3: Done
Dear Editor, With due all respect, forgives us Sorry for my oversight in the introduction part, we go directly to our aim.
we appreciate your support with provide us with fulfillment paper references and advice. we did the correction and added the references that will make the work more viable and clear and help the readers.
Point 4: The readers would also appreciate short information on different mechanism regulating melanin pigmentation (Physiol Rev 84, 1155-1228, 2004; Pigment Cell Melanoma Res 25, 14-27, 2012) and mentioning various properties of melanin under pathological and physiological conditions (Frontiers in Oncology 2022;12. DOI: 10.3389/fonc.2022.842496).
Response 4: Done
Point 5: Finally, the readers would appreciate histological figures with the description.
Response 5: Not applicable
Dear revieweer, I agree with you, unfortunately, histopathology results do not apply to this pilot study,
that due to financial difficulties ( this Pilot study is self-financial work), other technical issues, and insufficient samples,
this Pilot is an introduction to our current work, we will apply histopathology in our current study to expand the study of cosmetic effects under levels of the skin, hair, nails, urine, and blood samples, with investigations, much biochemical analysis including, heavy metals, histopathology, blood films, enzymes, endocrine parameters, etc (in progress).
The references that you advised will help us also.

Reviewer 3 Report
The authors were interested in pomegranate seed oil extract (PSOE) in view of medicinal cosmetology. The authors showed the results using rats that phenobarbital- and formaldehyde-induced skin oxidative stress and inflammation were ameliorated by the treatment of PSOE, and compared with the results of human volunteers. The authors concluded PSOE is useful for local treatment. The study was well documented and discussed including the comparison with the historical uses of PSOE. However, the manuscript needs some corrections for publication.
Major points
1) The presentation of the data was redundant. All figures (bar chart) contain tables, but it is recommended the authors choose one of them (bar or table). Additionally, % change Figures (Figs. 2, 4, and 6) were not mandatory if the statistical analyses were appropriately done. Generally, data is expressed as bar charts with error bars, and *, or #, if the analyses meet the significant level set before the experiments.
2) The vertical axis should not be expressed as “Mean”. The authors should not place the data with different units in a panel. Hence, Figures 1, 3, and 6 should be subdivided into (A), (B), …, (G).
3) The color of the Figures should be unified.
4) The 3D decoration should be avoided in Figures 1, 2, 4, 5, and 6.
5) The readers need an introduction to Avocom-M and TROSYD, before the results. For example, the clinical usage, merits and demerits, and some other mandatory properties to read the following results.
Minor points (a typo)
Fig. 7 Stage 2 phe-nobarbital --> phenobarbital
Minor points (suggestion)
It is easy to read if Figure 7 is placed very after the Introduction.
Author Response
- Dear reviewer with due all respect, it is our honor to received your revision to our work, that gives it more valuable and clarity.
Point 1: The presentation of the data was redundant. All figures (bar chart) contain tables, but it is recommended the authors choose one of them (bar or table). Additionally, % change Figures (Figs. 2, 4, and 6) were not mandatory if the statistical analyses were appropriately done. Generally, data is expressed as bar charts with error bars, and *, or #, if the analyses meet the significant level set before the experiments.
Response 1: Done
- Dear reviewer, we agree with you, after your permission it is our intent to make the data more redundant, the readers (ex. undergraduate students, master students, ...etc) would appreciate the full description of the pilot study to be simple and clear, this way make a clear cascade to our result analysis.
- (All figures (bar chart) contain tables), hence this is the kind of figure output of the excel program it is simple we choose this kind of graph because some data are very small that do not appear in the bar.
- (% change Figures), it is one of our aims for experimental design, after your permission we wish to keep it as it is which may help some readers and this is make our result analysis in simple cascade category.
- And we also provided charts as (mean ± SD) Statistical significance was set to p ≤ 0.05 as you advise (we put it in materials and method and in the result).
Statistical significance was set to p ≤ 0.05; means reported with the same letters (a, b, c, d) were not significantly different.
- We will take your advice in our current study of large-scale in progress.
Point 2: The vertical axis should not be expressed as “Mean”. The authors should not place the data with different units in a panel. Hence, Figures 1, 3, and 6 should be subdivided into (A), (B), …, (G).
Response 2: Done
Point 3: The color of the Figures should be unified.
Response 3: Done
The color unified is done according to the kind of study (human or animal), and also according to the type of discussion in the figure that will help readers to identify which of which.
Point 4: The 3D decoration should be avoided in Figures 1, 2, 4, 5, and 6.
Response 4: Done
Done in all, except for the percentage figures regarding clarity purposes.
Point 5: The readers need an introduction to Avocom-M and TROSYD, before the results. For example, the clinical usage, merits and demerits, and some other mandatory properties to read the following results.
Response 5: Done
Point 6: Minor points (a typo)
Response 6: Done
Point 7: Minor points (suggestion)
It is easy to read if Figure 7 is placed very after the Introduction.
Response 7: Done

Round 2
Reviewer 2 Report
The authors were responsive to the critique and adequately revised the manuscript
Author Response
Dear reviewer with due all respect, it is our honor to receive your revision to our work, which gives it more valuable and clarity.
Thank you.
Reviewer 3 Report
I understand all of the authors' explains and agree with the publication.
Author Response

(The authors gave the same response as above.)
